# Changes in Trunk Muscle Activity during Unilateral Weight Bearing and Abnormal Postural Gait in Healthy Individuals

**DOI:** 10.3390/medicina58121800

**Published:** 2022-12-06

**Authors:** Sungwoo Paek, Jungjoong Kang, Bokyung Shin, Jiyoon Jung, Hanee Rim, Mijeong Yoon, Kyoungbo Lee, Yeunjie Yoo, Boyoung Hong, Seonghoon Lim, Joonsung Kim

**Affiliations:** 1Department of Rehabilitation Medicine, St. Vincent’s Hospital, College of Medicine, The Catholic University of Korea, Seoul 06591, Republic of Korea; 2Department of Rehabilitation Medicine, Booboo Medical Healthcare Hospital, Yeongsanro 1666, Mokpo 58655, Republic of Korea

**Keywords:** torso, muscle, electromyography, weight-bearing, gait

## Abstract

*Background and Objectives:* Many people tend to carry their bags or baggage on only one side of their body. Due to smartphone use, people also tend to walk bent forward in a kyphotic posture. In this study, we aimed to assess trunk muscle activity changes due to weight-bearing, carried in the left or right hand, and using three different gait postures. *Materials and Methods:* We recruited 27 healthy participants (aged 19–75 years) with no history of LBP within the last 6 months before study participation. Electromyographic activities of the lower back and the abdominal muscles of the participants were evaluated using four-channel surface electromyography (EMG). Surface EMG recordings were obtained from four trunk muscles, including the flexor (rectus abdominis (RA), external oblique (EO)) and extensor muscles (lumbar erector spinae (LE), and the superficial lumbar multifidus (LM)), during unilateral weight-bearing tasks and with different gait postures (normal gait, with a sway back, and thoracic kyphosis). *Results:* In the “unilateral weight-bearing task”, there was a significant difference in the activity of all the trunk muscles between the weight-bearing limb side and the opposite side (*p* < 0.05). The activation of the left trunk muscle was greater than that of the right trunk muscle when the dumbbell was lifted using the right hand. The other side showed the same result. In the “gait posture task” performed by the participants using a sway-back posture, the RA and EO had a higher level of activity in the stance and swing phases compared with that in a neutral gait (*p* < 0.05). Moreover, in the participants with a thoracic kyphosis posture, the LE and LM had a higher level of activity compared with that in a neutral gait (*p* < 0.05). *Conclusions:* Our results indicate that abnormal gait posture and unilateral weight-bearing tasks may impair the balance of trunk muscles, increasing the incidence of LBP. However, further large-scale, prospective, controlled studies are warranted to corroborate our results.

## 1. Introduction

Walking is an essential human ability, and it is mandatory for social activities. Its health benefits are widely advocated for both young and older adults. Walking is often incorporated into rehabilitation programs [1].

Walking is mainly controlled by both the leg and hip muscles, and the whole body is involved in locomotion. Moreover, walking is also characterized by opposite arm swings and rotational movements of the trunk [2]. These complex activities involve both the stability and mobility of the vertebral column and, together with the joints and ligaments, the trunk and back muscles ensure the flexibility and integrity of the spine [3].

In humans, the trunk accounts for 60% of the total body mass, and its higher location relative to the lower limbs produces an “inverted pendulum” leverage effect [4]. Its versatility when participating in diverse types of motor activities arises from its connections to the muscles and joints [5]. It would, therefore, be mandatory to regulate the trunk movements resulting from muscular actions since this is essential for ensuring the above versatility [4].

Trunk stability is an essential element of the balance and coordinated use of the extremities in the various activities of daily living [6]. This is because the trunk plays a pivotal role in coordinating the body’s movements: proximal trunk control is mandatory for movement control, balance, and the functional mobility of the distal limb [7]. Therefore, trunk stability equips the trunk muscles with the ability to allow the body to remain upright, adjust for weight shifts, and perform selected movements. This eventually contributes to maintaining the center of mass within the base of support during static and dynamic postural adjustments [8]. Thus, trunk control serves as an integral part of the functional walking gait; healthy individuals maintain the trunk in a neutral orientation, with negligible excursions on the sagittal, coronal, and transverse planes during the gait [9,10,11]. This is because the trunk muscle actively contributes to balance during functional activities [4]. Once weakened, the trunk muscle interrupts posture control and compromises spinal stability. This eventually leads to functional impairment [12,13].

Bergmark [14] proposed that there are two muscle groups that control the trunk. The first group includes those muscles attached directly to the lumbar vertebrae that can provide spine segmental stability. The lumbar multifidus, transversus abdominis, and internal oblique (IO) muscles constitute this group. The second group consists of large torque-producing muscles with no segmental attachment to the lumbar spine. The muscles in the second group control the gross trunk movement and provide general trunk stability. They include the rectus abdominis (RA), external oblique (EO), and thoracic erector spinae muscles [15,16].

Moreover, a previous biomechanical study explored the involvement of abdominal muscles in the dynamic stability of the spinal vertebrae and reported that the IO was the most important muscle, followed by the EO and RA [17].

The lumbar erector spinae (LE) has shorter sagittal lever arms compared with those of the multifidus muscles. Unilateral activity gives the function of extension and lateral bending of the lumbar spine, and bilateral action extends the lumbar spine and stabilizes the spine, mainly in the lateral direction [14].

Trunk stability is important in functional walking and the activities of daily living. Currently, many individuals tend to carry their bags or baggage on one side only. Thus, there is a high possibility of trunk muscle imbalance, which can increase the incidence of low back pain (LBP). Unilateral weight-bearing tasks were conducted to assess whether a right-to-left side imbalance of trunk muscles occurs when carrying objects on one side, and how the activity of the right and left trunk muscles differs. Despite several studies evaluating trunk muscle imbalance to date, none has evaluated the changes in trunk muscles caused by abnormal posture during walking. In this study, we investigated changes in trunk muscle activity in the function of different abnormal postures.

## 2. Materials and Methods

### 2.1. Study Methods

#### 2.1.1. Subjects

This 1-day trial included healthy adults aged 19–75 years who had no history of LBP within the last 6 months before study participation. We excluded persons with a past history of LBP; any underlying disease, including infections, rheumatoid diseases, neurological deficits, spinal fractures, spinal deformity, scoliosis, tumors, or radiating pain; implanted electronic devices; body mass index ≥ 30 kg/m^2^; and a Cobb angle > 10°.

All participants were checked for underlying diseases and surgical history telephonically before visiting the hospital for examination. When the participant visited the hospital on the day of the surface electromyography (EMG) examination, the surgical history, underlying diseases, height, and weight of each participant were checked in an interview, and only healthy participants without any medical issues underwent the examination.

Sample size estimation was based on the central limit theorem: the distribution of a sample variable approximates a normal distribution (i.e., a “bell curve”) as the sample size becomes larger, assuming that all samples are identical in size, regardless of the population’s actual distribution shape. Sample sizes of ≥30 are often considered sufficient for the central limit theorem to hold. Thus, we aimed to recruit > 30 healthy participants, but 27 were finally recruited due to a lack of research funds.

We finally enrolled a total of 27 participants for this study. The study was approved by the Institutional Review Board of St. Vincent’s Hospital (IRB approval VC21EISI0054). The study was conducted in compliance with the relevant ethics guidelines. All the procedures described herein were performed in accordance with the 1964 Declaration of Helsinki and its later amendments or with comparable ethical standards.

#### 2.1.2. Baseline Assessment

Participant baseline characteristics, including underlying disease, age, gender, weight, and height, were evaluated.

#### 2.1.3. EMG Protocol

To reduce skin resistance prior to the attachment of an electrode, the skin on the target site was disinfected using alcohol, if necessary, and then shaved.

We chose two flexor muscles (RA and EO) and two extensor muscles (LE and the superficial lumbar multifidus (LM)), which provide spine segmental stability. These are the most commonly used muscles, based on previous research. Moreover, structurally speaking, electrodes can easily be attached to these muscles [1,15,18]. Each surface electrode was attached parallel to the muscle fibers. Each electrode was fixed with a tape and excessive mobilization was prevented accordingly. The locations of the placement of electrodes were as follows: RA, 5 cm inferior to the xiphoid process and 3 cm lateral to the midline; LE, L1 level, between the midline and lateral aspect of the body at the L1 vertebral level; LO, midway between the anterior superior iliac spine and rib cage, parallel to the muscle fibers; and LM, at 2 cm lateral to the midline, in the L4–L5 interspinous space (Figure 1).

The electromyographic activities of the lower back and abdominal muscles of the participants were analyzed using a 4-channel electromyogram (BTS FreeEMG 1000; BTS Bioengineering, Milan, Italy).

Gait was analyzed using G-Studio (G-Studio software, BTS Bioengineering, Milan, Italy). The duration of the stance and swing phases were quantified.

#### 2.1.4. Unilateral Weight-Bearing Tasks

The task was performed by alternatively holding a dumbbell on one side of the body, twice on the right side, and twice on the left side (Figure 2). The weights of the dumbbells were 6 and 4 kg in men and women, respectively. The weight of the dumbbells was considered with reference to the fact that the average weight of a woman’s handbag was 3.03 kg. For men, we decided to use a 6 kg dumbbell because we thought that men needed a similar stimulus with a slightly heavier weight than that for women. While they were holding the dumbbell, we instructed the participants to look straight ahead and keep the trunk in a straight posture. The activity of trunk muscles was measured in association with the weight of the dumbbell, as previously described [19,20].

#### 2.1.5. Electromyographic Activities of Four Trunk Muscles, Depending on Gait Posture

The participants were also evaluated for changes in the electromyographic activities of the four trunk muscles, depending on gait posture, while walking on a treadmill [1,18].

Many previous studies have used a human average walking speed of 4 km/h; however, this speed was thought to be dangerous when we tested walking in kyphotic and sway-back postures on the treadmill. Therefore, the walking speed was determined to be 3 km/h, within a non-dangerous range [21].

We defined erect standing, sway standing (sway back), and thoracic kyphosis as follows. The angle between the markers was measured using an electronic goniometer.
Erect standing: a posture that is characterized by the formation of an angle of 180° between the markers attached to the acromion, greater trochanter, and lateral malleolus [15].Sway standing (sway back): a posture that is characterized by a difference of an angle of >15° between the markers attached to the acromion, greater trochanter, and lateral malleolus, compared with the erect standing posture, in such a condition that the pelvis is displaced anterior to the trunk in a relaxed state [15].Thoracic kyphosis: a posture that is characterized by a difference of an angle of >15° between the markers attached to the acromion, greater trochanter, and lateral malleolus compared with the erect standing posture, in such a condition that the pelvis is displaced posterior to the trunk in a relaxed state [15] (Figure 3).

### 2.2. Processing of Electromyographic Signals

The amplitude of the electromyographic signal was calculated as the mean of the root mean square values. Then, the total summation of the motor unit action potentials generated at a given electrode location during the contraction of trunk muscles was obtained. The EMG values were normalized: the values of electromyographic signals were expressed as the percentage of the activity of trunk muscles during a calibrated test contraction [22]. It would, therefore, be mandatory to provide a standard value for the normalization of raw data on the activity of trunk muscles.

Two standard values for normalization can be measured as reference values. The maximum voluntary isometric contraction (MVC) or the reference voluntary contraction (RVC) of a muscle, which needs relatively submaximal contraction, can be measured as reference values [23,24].

To measure MVC, participants need to adopt a position that causes the maximum muscle contraction. Even in healthy participants, it could be difficult to achieve maximum contraction; thus, we set the RVC as the reference value, which is relatively easy to measure. The EMG data values are normalized by dividing by the RVC. The crook lying double-leg raise was used to measure the RVC of the flexor muscles (RA, EO), while the prone lying position was used for that of the extensor muscles (LM, LES) [24,25].

### 2.3. Evaluation and Criteria

We analyzed differences in electromyographic activities of the four trunk muscles between right and left in the weight-bearing positions, using the following formulae when there was weight-bearing on the right and left sides, respectively:Distribution of electromyographic activities on the right side (%) = (Electromyographic activities on the right side)/(total electromyographic activities)
where the total electromyographic activities are the sum of electromyographic activities on the right side and those on the left side.

### 2.4. Statistical Analysis

All data are expressed as mean ± standard deviation or the number of participants, with a percentage, where appropriate. Statistical analyses were performed using SPSS, version 18, for Windows (SPSS Inc., Chicago, IL, USA). We analyzed the statistical significance of the differences in measurements, using the paired *t*-test and Mann–Whitney U test. Statistical significance was set at *p* < 0.05. 

When we analyzed the muscle activity for each gait posture (neutral, sway back, and thoracic kyphosis), normality was tested using the Kolmogorov–Smirnov test and the Shapiro–Wilk test. The measured muscle activity for each group did not satisfy normality. Thus, for comparisons of independent groups that did not satisfy normality, we paired groups and compared the “neutral–sway back” groups and the “neutral–thoracic kyphosis” groups. We used the Mann–Whitney U test, a non-parametric statistical test, to compare two samples or groups.

## 3. Results

### 3.1. Baseline Characteristics of the Participants

A total of 27 normal healthy participants were enrolled in the current study: 11 men (40.7%) and 16 women (59.3%); mean age, 26.5 ± 7.4 (range, 18–45) years. The baseline characteristics of the participants are presented in Table 1.

### 3.2. Activity of the Trunk Muscles during Unilateral Weight-Bearing Tasks

The paired *t*-test and effect size for the paired *t*-test (Cohen’s d) were calculated to compare the distribution (%) of the right-hand side trunk muscle activation. Furthermore, significant differences were shown in all the trunk muscles between the weight-bearing limb side and its opposite side (*p* < 0.05) except for the RA muscle in the neutral standing position and on WBL (weight-bearing with the left hand) (*p* = 0.050) (Table 2 and Figure 4). Although the *p*-value of this RA muscle was not less than 0.05, it was borderline statistically significant (*p* = 0.05).

When standing in the neutral standing position, not lifting any dumbbells, the distribution (%) of the right trunk muscle activation (RA, EO, LE, and LM) was between 47.2 ± 10.4 and 50.6 ± 5.0, with little difference between the right and left sides (Table 2 and Figure 4).

After right-hand side weight-bearing by lifting the dumbbell using the right hand, the distribution of the right trunk muscle was calculated, and the value was between 34.5 ± 11.2 and 46.0 ± 6.5. This result means that the distribution of the left trunk muscle was relatively high. It was found that the activation of the left trunk muscle was greater than that of the right when the dumbbell was lifted using the right hand (Table 2 and Figure 4).

After left-hand side weight bearing by lifting the dumbbell using the left hand, the distribution of the right trunk muscle was calculated, and the value was between 51.4 ± 7.0 and 68.2 ± 11.6. This result means that the distribution of the right trunk muscle was relatively high. We found that the activation of the right trunk muscle was greater than that of the left when the dumbbell was lifted using the left hand.

Conventionally, *t*-test effect size values of 0.2, 0.5, and 0.8 are considered small, medium, and large, respectively [26]. According to this rule, the EO and LE muscles showed large effect-size values (Table 2; d > 0.8).

### 3.3. Activity of the Trunk Muscle Depending on Gait Posture

The gait phase of the EMG data consisted of the swing and stance phases. For comparisons between data groups without a normal distribution, a non-parametric test, the Mann–Whitney U test, was used. Normal gait (neutral position), sway back, and thoracic kyphosis were compared (Table 3 and Table 4 and Figure 5 and Figure 6).

In the patients demonstrating the sway-back posture, the RA and EO had a higher level of activity in the stance and swing phases compared with that in neutral gait (*p* < 0.05). Moreover, in the patients with thoracic kyphosis posture, the LE and LM had a higher level of activity compared with that in a neutral gait (*p* < 0.05) (Table 3 and Figure 5 and Figure 6).

## 4. Discussion

In gait movements, the trunk muscles play a crucial role in generating and regulating the motion between the trunk and pelvis [2,21]. Trunk muscles also play a role in balancing the trunk on the pelvis [27].

Trunk muscles are also an essential element, forming the complex interworking structures of the lumbar spine. Thus, they are involved not only in maintaining the stability of the vertebral column but also in controlling intervertebral spinal motion. This contributes to maintaining the stability of the lumbar spine, as well as the mobility of the human body [28,29,30].

It has been reported that unilateral loads cause changes in plantar pressure, as well as increased body energy consumption and EMG patterns in both limbs [31,32]. In this study, as expected, our results showed that there was no significant difference in the activity of trunk muscles between the left and right sides and between the front and back in the neutral posture. However, the activity of trunk muscles was relatively higher on the opposite side of the weight-bearing limb. This is in agreement with previous studies, showing that a difference in muscle activity could be detected, depending on the posture or weight-bearing activity [18,33]. In addition, the size effects of the EO and LE muscles were greater than 0.8 (Table 2), indicating that these muscles are relatively more easily affected during “unilateral weight bearing” than other muscles. Based on these results, it is conceivable to assume that prolonged abnormal posture leads to an imbalance in the EO and LE muscles. Muscle imbalance has been identified in LBP patients [31]; thus, it is conceivable that low back pain may be more likely to occur in the EO and LE muscles. In addition, we found that the LE and RA showed the greatest and smallest changes in activity, respectively, during the unilateral weight-bearing tasks. We also uncovered differences in the activity of the LE, EO, LM, and RA, in decreasing order, during the unilateral weight-bearing tasks. This might be due to the structural and functional differences between the four trunk muscles.

Moreover, our results showed that there were changes in the activity of trunk muscles, depending on gait posture. This suggests that there might be a difference in the type of stress between the abdomen and lumbar area when there is a change in gait posture [15,18].

For participants with a sway-back posture, the %RVC of the flexor muscles (RA and EO) had a higher level of activity in the stance and swing phases, compared with that in a neutral gait (*p* < 0.05). Furthermore, in participants adopting the thoracic kyphosis posture, the %RVC of the extensor muscles (LE and LM) had a higher level of activity compared with that in the neutral gait (*p* < 0.05). We found that there was an increase in the retroversion of the trunk and pelvic muscles in the sway-back posture, compared with that in the neutral posture. This is in agreement with the findings of a previous study [15]. The results of this study show that various postural changes induced by different spinopelvic curvatures influence the activity of the trunk muscles during walking. However, there is a paucity of data supporting a relationship between postural changes and the muscular activity of the hip joint. However, previous studies have reported that the retroversion of the trunk increases the internal moment for flexion of the trunk, as well as that of the hip joint [18,34].

Our results cannot be generalized because the study has several limitations. First, we enrolled a small number of participants in the study. Thus, further large-scale studies are warranted. Second, we selected only four trunk muscles for the current analysis. Third, most of the participants were in their 20s (mean age, 26.48 ± 7.35). This may have caused a selection bias.

## 5. Conclusions

Many people tend to carry their bags or baggage on one side only. Due to the development of smartphones, many people also walk while bent forward in a kyphotic posture. We conducted this study to assess changes in the activity of trunk muscles during unilateral weight-bearing tasks and different gait postures in healthy Korean individuals. We concluded that postural abnormalities could cause muscle imbalance from side to side and from anterior to posterior. This muscle imbalance may increase the incidence of LBP in the future. Therefore, it is important for clinicians to present not only primary pain treatment, such as medication or physical therapy, but also fundamental alternatives, such as daily lifestyle and walking posture correction, for patients who complain of LBP.

In conclusion, our results indicate that deviations from both normal gait and unilateral weight-bearing tasks might impair the activity of trunk muscles. However, further large-scale, prospective, controlled studies are warranted to corroborate our results. In addition, the participants of this study were healthy adult men with no low back pain or spine deformity. Further study is warranted to ascertain whether similar changes in muscle activity can be observed in patients with low back pain.

## Figures and Tables

**Figure 1 medicina-58-01800-f001:**
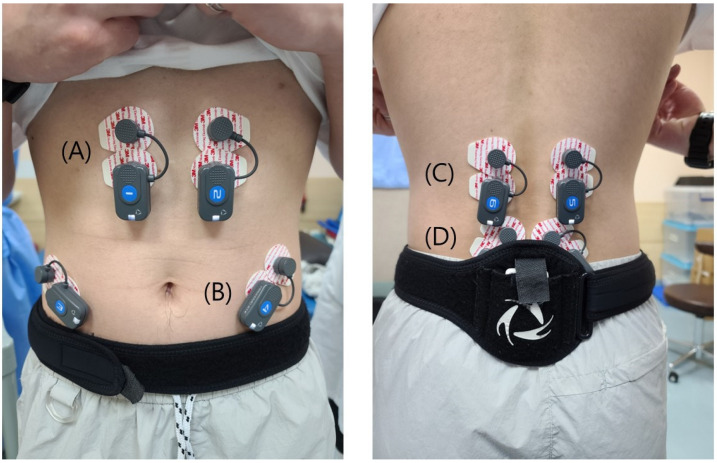
Locations of the placement of electrodes: (**A**) rectus abdominis; (**B**) external oblique. (**C**) lumbar erector spinae. (**D**) superficial lumbar multifidus.

**Figure 2 medicina-58-01800-f002:**
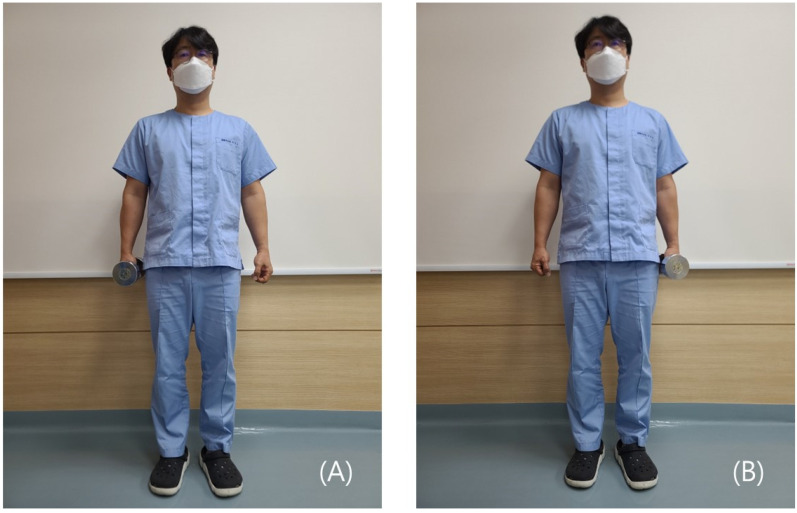
Unilateral weight-bearing tasks. (**A**) Weight-bearing on the right side. (**B**) Weight-bearing on the left side.

**Figure 3 medicina-58-01800-f003:**
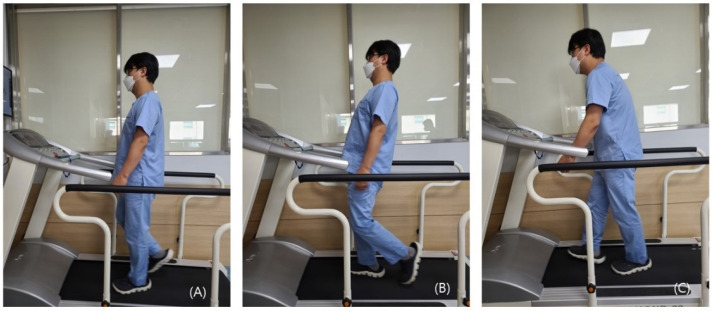
Three different gait postures: (**A**) normal gait; (**B**) sway back; (**C**) thoracic kyphosis.

**Figure 4 medicina-58-01800-f004:**
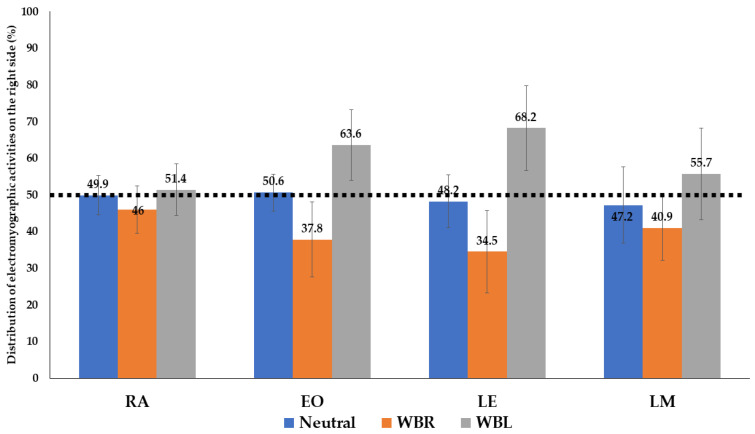
Distribution of electromyographic activities on the right side (%). After right-hand side weight-bearing activity, conducted by lifting the dumbbell (male, 6 kg dumbbell; female, 4 kg dumbbell) using the right hand (orange), the distribution of the right trunk muscle was lower than that of the neutral standing position (blue). After the left-hand side weight-bearing activity, conducted by lifting the dumbbell using the left hand (gray), the distribution of the right trunk muscle was higher than that in the neutral standing position (blue). Abbreviations: RA, rectus abdominis; EO, external oblique; LE, lumbar erector spinae; LM, superficial lumbar multifidus; WBR, weight-bearing with the right hand; WBL, weight-bearing with the left hand.

**Figure 5 medicina-58-01800-f005:**
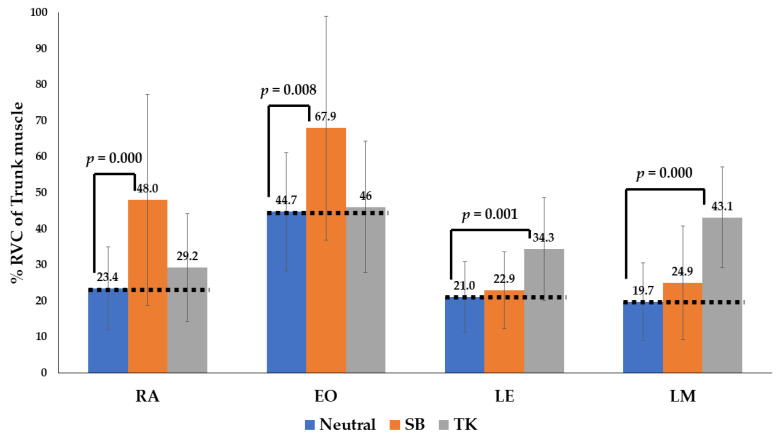
The percentage reference voluntary contraction (%RVC) of four trunk muscles during the swing phase of the gait. With the sway-back posture (orange), the RA and EO have a higher level of activity in the stance and swing phases compared with that of the neutral gait (blue). With the thoracic kyphosis posture (gray), the LE and LM have a higher level of activity than that of the neutral gait (blue). Abbreviations: SW, sway back; TK, thoracic kyphosis.

**Figure 6 medicina-58-01800-f006:**
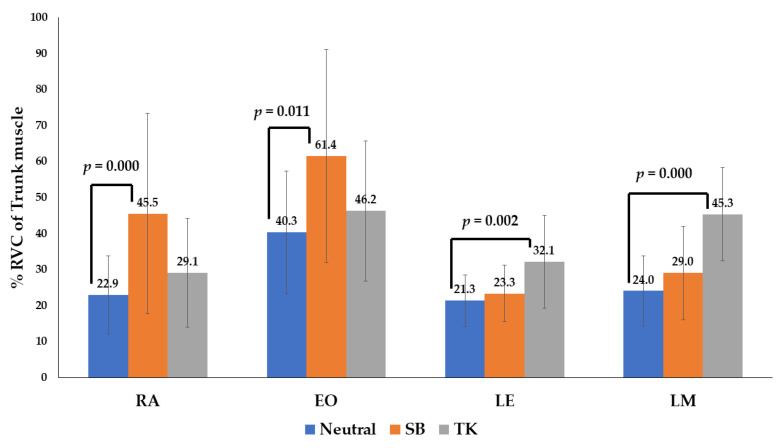
The percentage reference voluntary contraction (RVC) of four trunk muscles during the stance phase of gait. With the sway-back posture (orange), the RA and EO have a higher level of activity in the stance and swing phases compared with that of the neutral gait (blue). With thoracic kyphosis posture (gray), the LE and LM have a higher level of activity compared with that of the neutral gait (blue). Abbreviations: SW, sway back; TK, thoracic kyphosis.

**Table 1 medicina-58-01800-t001:** Baseline characteristics of the participants (n = 27).

Variables	Values
Age (years)	26.5 ± 7.4 (18–45)
Sex (male-to-female ratio)	11:16
Height (cm)	166.4 ± 8.1 (155–187)
Weight (kg)	61.4 ± 12.5 (46–86)

Values represent mean ± standard deviation with range.

**Table 2 medicina-58-01800-t002:** Comparison of trunk muscle activity, depending on the weight borne by each hand, using the paired *t*-test (male, 6 kg dumbbell; female, 4 kg dumbbell) (*n* = 27).

Rt. Distribution (%)							
Muscle	Neutral	Neutral—WBR			Neutral—WBL		
		WBR	*p*	Cohen’s d	WBL	*p*	Cohen’s d
RA	49.9 ± 5.3	46.0 ± 6.5	0.002 *	0.66	51.4 ± 7.0	0.050	0.24
EO	50.6 ± 5.0	37.8 ± 10.2	0.000 *	1.60	63.6 ± 9.6	0.001 *	1.70
LE	48.2 ± 7.2	34.5 ± 11.2	0.000 *	1.47	68.2 ± 11.6	0.000 *	2.06
LM	47.2 ± 10.4	40.9 ± 8.8	0.049 *	0.65	55.7 ± 12.5	0.002 *	0.74

Distribution of electromyographic activities on the right side (%) = (electromyographic activities on the right side)/(total electromyographic activities). Abbreviations: WBR; weight-bearing with the right hand, WBL; weight-bearing with the left hand. Statistical significance was set at *p* < 0.05, * *p* < 0.05.

**Table 3 medicina-58-01800-t003:** The percentage reference voluntary contraction (RVC) of four trunk muscles during the swing and stance phases of gait.

	Muscles	Mean %RVC		
		Normal	SB	TK
Swing Phase	RA	23.4 ± 11.5	48.0 ± 29.3 *	29.2 ± 15.0
	EO	44.7 ± 16.4	67.9 ± 31.1 *	46.0 ± 18.2
	LE	21.0 ± 9.8	22.9 ± 10.6	34.3 ± 14.3 *
	LM	19.7 ± 10.7	24.9 ± 15.8	43.1 ± 14.0 *
Stance Phase	RA	22.9 ± 10.9	45.5 ± 27.8 *	29.1 ± 15.1
	EO	40.3 ± 16.9	61.4 ± 29.6 *	46.2 ± 19.5
	LE	21.3 ± 7.2	23.2 ± 7.9	32.1 ± 12.9 *
	LM	24.0 ± 9.8	29.0 ± 13.0	45.3 ± 13.0 *

Abbreviations: RVC, reference voluntary contraction; RA, rectus abdominis; EO, external oblique; LE, lumbar erector spinae; LM, superficial lumbar multifidus; SB, sway back; TK, thoracic kyphosis. (Number) * indicate statistical significance at *p* < 0.05.

**Table 4 medicina-58-01800-t004:** Comparison of the activity of four trunk muscles according to gait posture, analyzed by the Mann-Whitney U test (*n* = 27).

Variables	Group	Mean %RVC	Mean Rank	Sum	U	*p*-Value
RA_swing	Neutral	23.4 ± 11.5	16.4	378.0		
	SB	48.0 ± 29.3	30.6	703.0	102.000	0.000 *
	TK	29.2 ± 15.0	26.2	602.0	203.000	0.177
RA_stance	Neutral	22.9 ± 10.9	16.3	375.0		
	SB	45.5 ± 27.8	30.7	706.0	99.000	0.000 *
	TK	29.1 ± 15.1	26.2	603.0	202.000	0.170
EO_swing	Neutral	44.7 ± 16.4	18.3	420.0		
	SB	67.9 ± 31.1	28.8	661.0	144.000	0.008 *
	TK	46.0 ± 18.2	23.7	545.0	260.000	0.921
EO_stance	Neutral	40.3 ± 16.9	18.5	425.0		
	SB	61.4 ± 29.6	28.5	656.0	179.000	0.011 *
	TK	46.2 ± 19.5	25.4	584.0	221.000	0.339
LE_swing	Neutral	21.0 ± 9.8	22.0	505.0		
	SB	22.9 ± 10.6	25.0	576.0	229.000	0.435
	TK	34.3 ± 14.3	30.1	693.0	112.000	0.001 *
LE_stance	Neutral	21.3 ± 7.2	21.9	504.0		
	SB	23.3 ± 7.9	25.1	577.0	228.000	0.423
	TK	32.1 ± 12.9	29.6	681.0	124.000	0.002 *
LM_swing	Neutral	19.7 ± 10.7	21.2	488.5		
	SB	24.9 ± 15.8	25.8	592.5	212.500	0.253
	TK	43.1 ± 14.0	33.0	760.0	45.000	0.000 *
LM_stance	Neutral	24.0 ± 9.8	20.9	481.0		
	SB	29.0 ± 13.0	26.1	600.0	205.000	0.191
	TK	45.3 ± 13.0	32.6	750.0	55.000	0.000 *

Abbreviations: RVC, reference voluntary contraction; RA, rectus abdominis; LE, lumbar erector spinae; LM, superficial lumbar multifidus; EO; external oblique. Values represent mean ± standard deviation. * Statistical significance was set at *p* < 0.05, * *p* < 0.05.

## Data Availability

The data presented in this study are available upon reasonable request from the corresponding author. The data are not publicly available because of privacy concerns.

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
