# Peer review of "Changes in Trunk Muscle Activity during Unilateral Weight Bearing and Abnormal Postural Gait in Healthy Individuals"

_medicina, 2022, doi:10.3390/medicina58121800_

Round 1

Reviewer 1 Report

1. The introduction section lacks sufficient detail on why this study was conducted. What is the relevance of uni-lateral weight bearing task? What the authors wrote in the conclusion section seems more appropriate for the introduction.

2. This study seems to be a prospective study with an intervention on humans. I don't think consent can be waivered in this kind of study. Please be specific about why the consent was waived. 

3. Please provide rationale for sample size. Common readers are not familiar with central limit theroem and this should be described. 

4. The authors only focused on the spine deformity for the inclusion/exclusion of the cohort. However, in this kind of study, I believe any deformity or previous operation history on the lower leg should be also excluded. Also please provide how was the cohort recruited as this is not specified in the manuscript. 

5. Please be specific about how the muscles were selected. The authors mentioned several references but this should be described in more detail. 

6. In the electromyographic protocol section, full term of  LA and MA is not disclosed. Also, in this section why is the gait analyzed? Is this part of the protocol? If so, please provide the results and its relevance. 

7. In unilateral weight bearing tasks section : it is unclear how this task was performed. Was this done by alternatively holding dumbell on one side? If the dumbells were hold in both hands, what was the posture of the subject that made this tasks unilateral weight bearing?

8. Please be more specific about statistics. Why was Mann Whitney U test used? If the authors intention was to compare pre vs post weight bearing, then this seem to be wrong method. 

9. Figures are missing so I can't make proper comments on the figures. 

10. Not that the purpose of the current study is " to assess changes in the activity of trunk muscles during unilateral weight-bearing tasks and depending on gait posture in Korean healthy individuals." This has to be answered in the conclusion. The current conclusion do not give a sufficient answer to the study question in the introduction section.

Table 1 : The authors excluded the underlying diseases so the table should not include this information. 

Figures are missing so I can't make a comment about the figures. 

Reviewer 2 Report

Title: The title of the article should be corrected (English language editing). 

Abstract: Background does not support the essence of the study, seems that it is Objective of the study. Description of the Participants should be moved from Results to Methods section. Results section does not reveal real results. Conclusion - is inadequate. 

Introduction does not reveal the problem, why the study was carried out.

Subject are described inconsistently. Authors counted sample size 28, but there are 27 study participants. 

In the Baseline assessment chapter authors write that they evaluated underlying disease, but in title they say - healthy individuals. Underlying diseases, in my opinion, should be included into exclusion criterias, and they do not need to be evaluated, just intereviewed. 

What does this abbreviation  mean? Has to be explained in the text.

An electrode was attached to each side of the site in parallel with muscle fibers - this is unclear sentence. 

The subjects were assessed for the measurement of electromyographic activities of four trunk muscles while holding a dumbbell in one each hand - this is unclear.

Is gait posture the same as body posture while walking?

It is unclear - walking in kyphotic and lordotic positions.

This is unclear - We defined erect standing, swing standing, and thoracic kyphosis as follows. Because % maximum voluntary isometric contraction was difficult to pose, we measured and the % reference voluntary contraction (RVC), instead

Results and discussion chapters need to be rewritten with English language specialist help, as it is complicated to read and understand. 

Conclusion does not support the objective of the study. 

Some references (1967, 1981, 1989, 1992) are quite old.

Round 2

Reviewer 2 Report

Dear authors, I have the only one suggestion to you - to include titles of vertical axis in Figures No. 4, 5, and 6. 

Author Response

Thank you for your valuable comment. We have modified Figures 4, 5, and 6.